# Impact of Rotating Shifts on Lifestyle Patterns and Perceived Stress among Nurses: A Cross-Sectional Study

**DOI:** 10.3390/ijerph19095235

**Published:** 2022-04-26

**Authors:** Shang-Lin Chiang, Li-Chi Chiang, Wen-Chii Tzeng, Meei-Shyuan Lee, Chan-Chuan Fang, Chueh-Ho Lin, Chia-Huei Lin

**Affiliations:** 1School of Medicine, National Defense Medical Center, Taipei 11490, Taiwan; andyyy520@yahoo.com.tw; 2Department of Physical Medicine and Rehabilitation, Tri-Service General Hospital, Taipei 11490, Taiwan; 3School of Nursing, National Defense Medical Center, Taipei 11490, Taiwan; lichichiang@gmail.com (L.-C.C.); wctzeng@mail.ndmctsgh.edu.tw (W.-C.T.); 4School of Nursing, China Medical University, Taichung 40402, Taiwan; 5School of Public Health, National Defense Medical Center, Taipei 11490, Taiwan; meei.shyuan@msa.hinet.net; 6Department of Nursing, Yuan-Rung Hospital, Changhua 51045, Taiwan; jane_f56@yahoo.com.tw; 7Master Program in Long-Term Care, College of Nursing, Taipei Medical University, Taipei 110301, Taiwan; chueh.ho@tmu.edu.tw; 8Center for Nursing and Healthcare Research in Clinical Practice Application, Wan Fang Hospital, Taipei Medical University, Taipei 11031, Taiwan; 9Department of Nursing, Tri-Service General Hospital, Taipei 11490, Taiwan

**Keywords:** lifestyle pattern, perceived stress, rotating shift, shift work, sleep pattern, sleep quality

## Abstract

Although rotating shifts have a negative health impact, their association with hospital nurses’ health risks remains controversial due to incomplete adjustment in lifestyle patterns and heterogeneity of work schedules. However, whether work schedule characteristics are associated with lifestyle patterns and perceived stress remains undetermined. We assessed the correlations of work schedule characteristics, lifestyle patterns, and perceived stress among hospital nurses. This cross-sectional study included 340 nurses from two hospitals. Final data from 329 nurses regarding work schedule characteristics, lifestyle patterns (physical activity, dietary behavior, and sleep pattern), and perceived stress were analyzed via linear regression models. Fixed-day-shift nurses had reduced perceived stress (β = 0.15, *p* = 0.007) compared with rotating-shift nurses. Additionally, among rotating-shift nurses, fixed-evening- and fixed-night-shift nurses had longer sleep duration (β = 0.27, *p* < 0.001; β = 0.25, *p* < 0.001) compared to non-fixed-rotating-shift nurses. Longer rotating-shift work was associated with healthier dietary behaviors (β = 0.15, *p* = 0.008), better sleep quality (β = −0.17, p = 0.003), lower perceived stress (β = −0.24, *p* < 0.001), and shorter sleep duration (β = −0.17, *p* = 0.003). Hospital nurses’ work schedule characteristics were associated with lifestyle patterns, dietary behavior, sleep pattern, and perceived stress. Fixed-shifts were beneficial for lifestyle and lower perceived stress. Longer rotating shifts could help nurses adjust their lifestyles accordingly.

## 1. Introduction

Rotating shift work is related to significant health risks, including cardiometabolic diseases and cancer [1,2,3,4,5]. However, its association with health risks remains controversial, due to the heterogeneity of shift work schedules and incomplete adjustments for potential covariates, particularly lifestyle patterns.

Previous studies identified an association between rotating shifts and an increased risk of cardiometabolic, and gastrointestinal diseases and cancer-related mortality among hospital nurses [2,6,7]. These health problems appeared to converge around lifestyle patterns (physical activity and dietary behaviors) and perceived stress [8,9]. In addition, reports regarding the impact of shift work on health outcomes are contradictory and mostly related to sleep patterns [10,11]. Incidences of cardiovascular disease, metabolic syndrome, and cancer were correlated with sleep quality and duration [12,13,14]. Substantial evidence demonstrated the association between sleep duration and health risks [15,16]. In addition, previous studies revealed that nurses, particularly those who worked night shifts, experienced poor sleep quality [11,17]. Therefore, confirming a causal relationship between rotating shifts and health risks without considering lifestyle covariates (i.e., physical activity, dietary behaviors, and sleep pattern) and perceived stress would result in bias.

To adapt to the needs of various periods, an internal regulatory mechanism, that changes with a 24-h cycle, adjusts physiological functions, such as the sleep–wake cycle and vital organ function, is required [18]. For rotating-shift workers, undertaking activities during their originally-scheduled sleep period may lead to the readjustment of their physiological systems [18]. However, whether it reaches a new constant or steady-state remains inconclusive [18]. Globally, high proportions of hospital nurses engage in shift work, resulting in the desynchronization of their circadian rhythms, which may cause potentially adverse health consequences, due to complex factors. Factors affecting circadian rhythms involve diverse external stimuli, such as light, physical activity, dietary behavior, and sleep pattern, and social stimuli (i.e., perceived stress). In addition to the physiological effect (circadian rhythm disruption), there may be behavioral and psychological effects of shift work exposure [19]. Therefore, exploring these combined effects on rotating shift work among hospital nurses can provide a comprehensive understanding of the regulation of their circadian rhythms and the association between shift work and health risks.

Lifestyle patterns are closely linked to health risks [12,14]. A recent cohort study signified that rotating shift work and unhealthy lifestyle patterns were associated with adverse health outcomes. Moreover, in this joint association, unhealthy lifestyle patterns were more strongly correlated to health risks than rotating shift work in female nurses, indicating that adoption of healthy lifestyle patterns could prevent health risks [2]. Rotating shifts influence an individual’s willingness to maintain healthy lifestyle patterns, such as active physical activity patterns and healthy dietary behaviors [2,20]. Unfortunately, insufficient physical activity, unhealthy dietary behavior, poor sleep pattern, and high perceived stress can cause deterioration in health [8,9]. Therefore, exploring the association between the characteristics of work schedules, lifestyle patterns, and perceived stress in rotating-shift nurses is crucial.

Hospital nurses tend to have unhealthy lifestyle patterns, such as low physical activity [21]. Factors that hinder their engagement in physical activity include their work itself and the traditional familial role played by most women in Asian cultures [22], particularly among those in rotating shifts. In Taiwan, females have a particularly high prevalence of low physical activity levels [23]. Considering that the majority of nursing professionals are female, it is important that studies focus on lifestyles of hospital nurses, especially those in rotating shifts [24]. Typically, rotating shifts reduce nurses’ willingness to be physically active, due to fatigue or sleeping problems, which makes them prefer static leisure activities [20]. However, an inconsistent finding by a recent six-year follow-up study indicated no differences in physical activity between nurses in different work schedules [25]. Therefore, evidence regarding the influence of work schedule on physical activity remains unclear.

Poor dietary behavior is a major behavioral factor that causes health risks. However, whether it is affected by work schedule remains inconclusive. Rotating-shift nurses have poor dietary behaviors [21], as they are distinct regarding preparing, choosing, or purchasing food, compared with those working normal day shifts [26]. Nevertheless, research on the dietary behaviors among rotating-shift nurses is insufficient and limited regarding cultural diversity [27]. Thus, we proposed that shift work (physiological effect of circadian rhythm disruption) combined with lifestyle patterns (behavioral effect of dietary behaviors) resulted in a model that explained health risks.

Another crucial factor among hospital nurses is perceived stress, which explains their high turnover rate. Furthermore, it is affirmed to be greater in hospital nurses compared with the general public and private enterprise employees [28]. Nurses reported higher perceived stress, closely linked to rotating shifts [29], which played a vital role in developing major non-communicable diseases [30]. In addition, as sleep quality deteriorates, and sleep duration becomes shorter, perceived stress and burnout increases significantly [28]. However, up till now, there have been limited discussions regarding the characteristics of work schedule and perceived stress among hospital nurses.

Therefore, this study intended to assess whether work schedule characteristics were associated with lifestyle patterns (physical activity, dietary behavior, and sleep pattern) and perceived stress among hospital nurses.

## 2. Materials and Methods

### 2.1. Design

This study employed a cross-sectional research design to explore the associations between the characteristics of work schedule (type of work schedule, rotating shift type, and the length of rotating shift work), lifestyle patterns (physical activity, dietary behavior, and sleep pattern), and perceived stress among hospital nurses.

### 2.2. Participants and Settings

Registered nurses, aged 20–65 years, were selected from a medical center and a regional hospital in Northern Taiwan between January and December 2018. A total of 1700 hospital nurses, who had been at least three months on-duty, were initially screened using employee records. Subsequently, a sample was obtained through stratified random sampling, with an equal percentage of 20% from 42 work units/specialty areas, via computer-generated serial numbers using SPSS version 16.0 (SPSS Inc., Chicago, IL, USA). This process, in tandem with eligibility and final enrollment, was supervised by the principal investigator. Finally, 340 randomly selected participants were invited for further assessment by a research assistant (RA) after their informed consent was obtained. The RA provided the paper questionnaires to each participant after they explained the research purpose and questionnaire to minimize answer bias. The completed questionnaires were returned to the head nurses of each work unit before being collected by the RA after two weeks. To increase response rate, participants were compensated with 7 USD after questionnaire completion.

The inclusion criteria were: (1) hospital nurses aged 20–65 years, (2) at least three months of work experience at the local hospital, (3) a full-time worker with a maximum of 46 h overtime per month, (4) could speak and understand Mandarin, and (5) consented to participate in the research. The exclusion criteria were: (1) had less than three months work experience at the local hospital, (2) part-time worker, (3) refused to participate or was unable to provide written informed consent, and (4) history of mental illness or cancer, which is closely linked to lifestyle changes [31], (5) experienced a major event (i.e., death of a loved family member) in the past six months, since it was related to perceived stress.

The sample size estimation based on a linear multiple regression (random model) with an estimated difference of 0.1, assumed 30 predictors/covariates, an alpha of 0.05, and a power of 0.8, which resulted in a total of 282 participants [32]. Considering an 85% estimated response rate, a sample of 332 hospital nurses was required.

### 2.3. Measures

Participants’ sociodemographic characteristics (age, gender, marital status, education, work unit/specialty area, and length of work), work schedule characteristics (work schedule type, rotating shift type, and length of rotating shift work), lifestyle patterns (physical activity, dietary behavior, and sleep pattern), and perceived stress were measured.

#### 2.3.1. Work Schedule Characteristics

This study used two main work schedule types (fixed-day vs. rotating shifts). A fixed-day shift was determined as working only the day-shift (08:00–16:00, 8 h) in the month. A rotating shift, working at least two shifts in a month, was classified into three rotating shift types: (a) non-fixed-rotating shift, (b) fixed-evening shift, and (c) fixed-night shift. A non-fixed-rotating shift was presented as working three different shifts (day [08:00–16:00, 8 h], evening [16:00–24:00, 8 h], and night [00:00–08:00, 8 h] shifts) in a month. There was no regular direction of shift rotation in non-fixed-rotating shift, which can be clockwise (i.e., day/evening/night) or counter-clockwise (i.e., night/evening/day) and all work schedules were without a quick return (<11 h) between two consecutive shifts. The length of each work schedule was no longer than 6 days. A fixed-evening or fixed-night shift was defined as working the same evening or night shift in a month for at least 16 days and undertaking at most two shifts. For example, a fixed-evening shift may comprise 16 days of evening shift and 4 days of day shift or 17 days of evening shift and 3 days of night shift. Additionally, the length of the rotating shift (the duration of rotating shifts experienced in years) was also assessed.

#### 2.3.2. Lifestyle Patterns

##### Physical Activity

The reliable and widely-valid Taiwanese version of the International Physical Activity Questionnaire-Short Form (IPAQ-SF) with seven items was used to estimate physical activity, which involved housework, transportation, and leisure activity for the past week [33]. The scores, expressed as metabolic equivalent (MET)-minutes/week, were calculated by multiplying MET level with activity events (walking: 3.3 METs, moderate-intensity: 4.0 METs; vigorous: 8.0 METs) and duration (minutes) per week [34]. The sum of all walking, moderate-intensity, and vigorous physical activities was presented as the individual’s weekly amount of physical activity. According to the recommendations of the Centers for Disease Control and the American College of Sports Medicine [35], weekly physical activity determined by the IPAQ scores can be categorized into three levels: low—less than 600 MET-minutes/week, moderate—at least 600 MET-minutes/week (three or more days a week of vigorous activity of at least 20 min per day or five or more days of moderate-intensity activity or walking for at least 30 min per day), or high—vigorous-intensity activity of at least three days a week and the accumulation of at least 1500 MET-minutes/week or seven days of any combination of walking, moderate, or vigorous activities that achieved a minimum of at least 3000 MET-minutes/week.

##### Dietary Behavior

The Taiwanese version of the 21-item Modified Dietary Behavior Scale, developed by Lu [33], was utilized to evaluate participants’ general dietary behaviors and nutritional intake behaviors (high sodium, fat and cholesterol, sugar, and fiber dietary behavior). This scale has been reported to show good reliability and validity [36]. Each item was scored from 1–5, and the range of the total score was 21–105. Higher scores indicate healthier dietary behaviors among participants.

##### Sleep Pattern

The Taiwanese version of the Pittsburgh Sleep Quality Index, originally developed by Buysse et al. [37], was applied to assess sleep pattern. This 19-item scale, with good validity and reliability, included personal subjective sleep quality, sleep duration, latency, efficiency, disturbances, daytime dysfunction, and use of medication [38]. The score for each category ranged from 0–3 points, and the total score ranged from 0–21 points. Higher scores indicated deteriorating sleep quality [35]. In addition, we analyzed the category of sleep duration (hours/day), since substantial evidence has proved the association between sleep duration and health risks [15,16].

#### 2.3.3. Perceived Stress

Perceived stress, subjective perception of social psychology, and environmental stressors, were assessed via the 10-item Chinese version of the Perceived Stress Scale. Its reliability and validity have been well established in various populations [39]. Each item was scored on a 5-point Likert scale that ranged from 0–4, and the total score ranged from 0–40. Higher scores indicated greater perceived stress.

### 2.4. Data Analysis

Statistical analyses were performed using SPSS version 16.0 (SPSS Inc., Chicago, IL, USA). Descriptive data analysis was presented as mean (standard deviation) and frequency (percentage). Inference data were analyzed via a chi-square test, *t*-test, analysis of variance, Pearson product difference correlation, and multilinear regression models. Preliminary analyses were conducted before the inferential statistics to describe the key features of the data and summarize the results for further analyses. Considering that sociodemographic factors were correlated with lifestyle patterns and perceived stress [40], this study appraised the association between work schedule characteristics, lifestyle patterns, and perceived stress under adjustment for those potential sociodemographic covariates via a multiple linear regression model, a statistical technique that used several explanatory (independent) variables to predict the outcome of a response (dependent) variable. In addition, dummy variables were created for analyses when an independent variable was categorical and had two or more distinct categories/levels, since regression analysis treats all independent variables as numerical. All statistical analyses were two-tailed and considered significant at *p* < 0.05.

## 3. Results

Although we initially recruited 340 participants, only 330 completed the study (97.1% response rate). However, only 329 participants were analyzed since one had missing information. The sociodemographic and work schedule characteristics of the participants are presented in Table 1. Table 2 illustrates their lifestyle patterns. There were no significant differences in physical activity, dietary behaviors, and sleep quality when lifestyle patterns between work schedule types (fixed-day and rotating shifts) were compared. However, fixed-day-shift nurses had a less healthy dietary behavior of high fiber consumption (54.5 vs. 63.5, *t* = −3.5, *p* = 0.001), shorter sleep duration (7.2 vs. 7.9, *t* = −4.27, *p* < 0.001), and lower perceived stress (17.3 vs. 18.8, *t* = −2.73, *p* = 0.007) compared with rotating-shift nurses.

Table 3 demonstrates that non-fixed-rotating-shift nurses had relatively less physical activity compared with fixed-evening- or fixed-night-shift nurses. Non-fixed-rotating-shift nurses reported less healthy dietary behaviors, particularly general dietary behavior, shorter sleep duration, lower sleep quality, and higher perceived stress compared with fixed-evening- or fixed-night-shift nurses. However, only general dietary behavior (*F* = 4.95, *p* = 0.008), salt consumption behavior (*F* = 3.46, *p* = 0.033), and sleep duration (*F* = 10.98, *p* < 0.001) demonstrated significant differences between the different types of rotating shifts. Post-hoc analyses using the Scheffe’s test further revealed that Non-fixed-rotating-shift nurses had less healthy general dietary behavior compared with fixed-night-shift nurses; fixed-night-shift nurses had less healthy dietary behavior of salt consumption compared with fixed-evening-shift nurses; and non-fixed-rotating-shift nurses had shorter sleep duration compared with fixed-evening- or fixed-night-shift nurses.

Table 4 presents the relationship between the length of rotating shift work and lifestyle patterns. A longer length of rotating shift work was associated with healthier dietary behaviors (*r* = 0.149, *p* = 0.008), decreased sleep duration (*r* = −0.169, *p* < 0.001), and reduced perceived stress (*r* =−0.238, *p* < 0.001).

Based on a multiple linear regression analysis (Table 5), after adjusting for sociodemographic covariates, work schedule type was found to be correlated with sleep duration (β = 0.18, *p* = 0.001) and perceived stress (β = 0.15, *p* = 0.007). Compared with fixed-day-shift nurses, rotating-shift nurses had longer sleep duration (β = 0.18, *p* = 0.001) and higher perceived stress (β = 0.15, *p* = 0.007). Likewise, regarding the association between rotating shift work and sleep duration, fixed-evening- and fixed-night-shift nurses had longer sleep duration (β = 0.27, *p* < 0.001; β = 0.25, *p* < 0.001) compared with non-fixed-rotating-shift nurses. Furthermore, the length of rotating shift work was associated with dietary behavior (β = 0.15, *p* = 0.008), sleep duration (β = −0.17, *p* = 0.003), sleep quality (β = −0.17, *p* = 0.003), and perceived stress (β = −0.24, *p* < 0.001). A longer length of rotating shift work was associated with healthier dietary behaviors, shorter sleep duration, and lower perceived stress among the participants.

## 4. Discussion

To the best of our knowledge, our study is the first to explore the association between work schedule characteristics, lifestyle patterns (physical activity, sleep pattern, and dietary behavior), and perceived stress. Specifically, we distinguished lifestyles based on the different types of work schedules to confirm which type of work schedule would be beneficial for the lifestyle of nurses. This study revealed that rotating shifts were associated with increased perceived stress compared with fixed-day shifts. Additionally, fixed-evening- or fixed-night-shift nurses had longer sleep duration compared with non-fixed-rotating-shift nurses. Participants with a longer length of rotating shift work reported lower sleep duration. However, they had healthy dietary behaviors, better sleep quality, and lower perceived stress.

Physical activity was a potential mediator of the association between rotating shifts and adverse health outcomes [19]. Our study utilized the IPAQ-SF, a validated and standardized tool, to compare the weekly amounts of physical activity among different work schedule types. Our results demonstrated that there were no differences in physical activity. These results were consistent with those of a previous study, which used accelerometers to measure physical activity in rotating-shift workers in comparison to fixed-day-shift workers [41]. Some studies reported that shift workers had significantly higher occupational physical activity compared with day workers [42,43], whereas others advocated that shift workers seldom engaged in occupational or leisure-time physical activity [21]. Factors that included work characteristics, types of physical activity (i.e., leisure-time physical activity, occupational physical activity, total physical activity, or sedentary behavior), measurement tools, and cultural differences yielded inconsistent conclusions [19,25]. Hence, future researchers should clarify the discrepancies using standardized, validated tools and elucidate whether specific physical activities contributed to adverse health consequences associated with rotating shift work among rotating-shift nurses.

Despite a lack of conclusive evidence regarding the correlation of rotating shift work and physical activity among rotating-shift nurses, possible barriers to engaging in regular physical activity have been assessed [20,22]. Rotating-shift nurses experienced difficulties in the implementation and maintenance of an active physical activity pattern, due to fatigue resulting from rotating shifts, an inability to participate in socially-formed physical activities or sports, and perceived exertion during exercise [20], along with complex perceived barriers [22].

Poor dietary behavior is a key lifestyle factor that causes non-communicable diseases [9]. Our study found that non-fixed-rotating-shift nurses had less healthy dietary behaviors compared with fixed-evening- or fixed-night-shift nurses, given that this behavior did not reach significance after adjusting for sociodemographic covariates. This result was consistent with that of a previous study, which stated that participants reported consumption of discretionary food due to limited availability of healthy food choices during night shifts [27]. Moreover, general dietary behaviors were less healthy among non-fixed-rotating-shift nurses compared with fixed-night-shift nurses. Non-fixed-rotating shifts can result in irregular eating patterns, which can disrupt the circadian rhythm. This may contribute to an unfavorable metabolic phenotype, thereby increasing the risk of chronic disease [19,21]. However, less healthy dietary behaviors, particularly high fiber consumption behavior, among the fixed-day-shift nurses of this study, may be due to the tightly scheduled routine following the day shift of hospitals in Taiwan, which forces nurses to consume fast food or discretionary foods (i.e., preferring handmade drinks over a full meal). This inclination may also be attributed to cultural factors. Additionally, as fixed-evening- or fixed-night-shift nurses tend to choose a low-quality diet [19], they may be familiar with work routines in specific scheduled shifts and can adopt healthier dietary behaviors gradually. The outcomes of our study affirm that longer rotating shift work was associated with healthier previously-followed dietary behaviors.

The impact of rotating shifts on increased health risks attributed to decreased sleep duration has received increased attention. Our study revealed that among nurses with rotating shifts, the fixed-evening- or fixed-night-shift nurses had longer sleep duration compared with non-fixed-rotating-shift nurses. Nonetheless, compared with fixed-day-shift nurses, rotating-shift nurses had longer sleep duration. Hence, due to unknown reasons, fixed-day-shift nurses did not sleep as long as rotating-shift nurses. Longer working hours in one day shift among fixed-day nurses compared with rotating-shift nurses and higher frequencies of the use of hypnotics (sleeping medicines) by rotating-shift nurses may explain this phenomenon [44]. In addition, although no significant differences in sleep quality were found between all work schedule types, our results revealed that fixed-evening-shift nurses had relatively better sleep quality compared with non-fixed-rotating-shift nurses. This finding implies that a fixed-evening or fixed-night shift is a favorable shift work type for hospital nurses to attain sufficient sleep. Furthermore, previous evidence advocated that short sleep duration predicted risk of cardiometabolic diseases [1,3]. In addition, this study found that a longer length of rotating shift work was associated with shorter sleep duration among hospital nurses. Nonetheless, length of rotating shift work was not associated with sleep quality. The reasons for this warrant further research.

Generally, hospitals create stressful work environments, and shift work places an additional strain on hospital nurses [29,42]. Therefore, the health risks of this population necessitate stress management and the transformation of the work environment to facilitate healthy lifestyles [21]. In this study, fixed-day-shift nurses had significantly reduced perceived stress compared with rotating-shift nurses. Among rotating-shift nurses, fixed-evening- or fixed-night-shift nurses had lower perceived stress than non-fixed-rotating-shift nurses. We postulated that a fixed shift schedule was beneficial to reduce perceived stress for hospital nurses, which could mitigate health risks related to stress. In addition, the longer the length of rotating shift work, the lower the perceived stress. This could be explained by the familiarity with work routines and accumulated competency regarding clinical emergency response, which helps hospital nurses cope with rotating shift-related stress gradually.

Hence, it is crucial for hospital employers or managers to be aware of the unhealthy lifestyles and psychological consequences of shift work, try to organize screening for nurses performing shift work, and design health promotion activities, or make resources available, for hospital nurses to engage in healthy lifestyle behaviors.

Several limitations should be considered. First, the data may be subject to recall bias as the information on rotating shift work, lifestyle patterns, and perceived stress were self-reported, which could potentially lead to misclassification. Second, as this was a cross-sectional design, causality could not be determined. Third, participants were recruited from only two hospitals in the same region in Taiwan and were relatively young with a mean age of 32.8 years. Hence, the geographic region may limit generalizability to other cultural groups. Fourth, a typical shift for a Taiwanese nurse was usually eight hours per day five days per week, which also involved working during weekends. Thus, the findings must be interpreted and compared with caution. Although we cannot neglect these limitations, the study had certain strengths, which included random sampling, a high response rate (96.8%), and adjusted socioeconomic covariates to examine the associations between work schedule characteristics and lifestyle patterns.

## 5. Conclusions

The characteristics of work schedule in hospital nurses were associated with lifestyle patterns, which involved dietary behavior, sleep pattern, and perceived stress. Early screening and lifestyle modification should be integrated into health promotion for rotating-shift nurses to prevent and reduce the negative effects of rotating shifts. Notably, a fixed shift work schedule or ergonomic turnaround schedule could be beneficial for healthy lifestyles and lower perceived stress, which may enhance hospital nurses’ holistic well-being.

## Figures and Tables

**Table 1 ijerph-19-05235-t001:** Demographic and work schedule characteristics of the participants (*N* = 329).

Variable	Mean ± SD	n (%)	Range
Age (*year*)	32.8 ± 8.6		21.2–61.6
Gender			
Male		15 (4.6)	
Female		314 (95.4)	
Education			
Associate degree		72 (21.9)	
Bachelor		242 (73.6)	
Master and higher		15 (4.6)	
Marriage			
Married		70 (21.3)	
Single		259 (78.7)	
Work unit/specialty area			
General medical/surgical ward		180 (54.7)	
Emergency/critical unit		113 (34.3)	
Other		36 (10.9)	
Length of work (*year*)	8.4 ± 8.2		0.3–40.0
Work schedule type ^a^			
Fixed-day shift		80 (24.3)	
Rotating shift ^b^		249 (75.7)	
Non-fixed-rotating shift		126 (38.3)	
Fixed-evening shift (*16:00*–*24:00*)		63 (19.1)	
Fixed-night shift (*00:00*–*08:00*)		60 (18.2)	
Length of rotating shift work (*year*)	7.6 ± 7.5		0.2–38.1

*Note*: Data were presented as mean ± SD or n (%); ^a^ Work schedule type included (1) fixed-day shift and (2) rotating shift; ^b^ Rotating shift was classified into (1) non-fixed-rotating shift, (2) fixed-evening shift, and (3) fixed-night shift.

**Table 2 ijerph-19-05235-t002:** Comparison of lifestyle patterns between the different work schedule types.

Variables	All Participants (*N* = 329)	Fixed-Day Shift (*n* = 80)	Rotating-Shift (*n* = 249)	*t*	*d*	*p*
*Physical activity*	1408.5 (1341.3)	1377.1 (1330.8)	1418.6 (1347.2)	−0.24	−0.03	0.81
*Dietary behavior*	66.0 (7.2)	65.8 (7.0)	66.1 (7.3)	−0.33	−0.04	0.74
General dietary behavior	62.0 (10.4)	61.4 (10.4)	62.1 (10.4)	−0.52	−0.07	0.60
Nutrient consumption behavior	63.5 (7.4)	63.5 (6.9)	63.5 (7.6)	0.01	0.001	0.99
Salt consumption behavior	66.2 (15.8)	67.0 (17.4)	66.0 (15.3)	0.50	0.06	0.62
High fat-density consumption behavior	63.3 (9.4)	63.9 (9.0)	63.1 (9.6)	0.61	0.08	0.54
Sugar consumption behavior	62.5 (13.0)	63.3 (13.6)	62.3 (12.9)	0.57	0.07	0.57
High fiber consumption behavior	61.3 (20.4)	54.5 (20.1)	63.5 (20.1)	−3.50	−0.45	0.001
*Sleep pattern*						
Sleep duration	7.7 (1.7)	7.2 (1.1)	7.9 (1.8)	−4.27	−0.55	<0.001
Sleep quality	7.2 (3.1)	7.0 (2.9)	7.2 (3.2)	−0.61	−0.08	0.54
*Perceived Stress*	18.4 (4.2)	17.3 (3.7)	18.8 (4.3)	−2.73	−0.12	0.007

*Note*: Data were presented as mean (SD); Physical activity (*MET*-*min*/*week*); Sleep duration (*hour*); *d*: standardized mean-difference effect size; *p* values from the Independent *t*-test.

**Table 3 ijerph-19-05235-t003:** Comparison of lifestyle patterns between the different types of rotating shifts (*n* = 249).

Variables	Non-Fixed-Rotating- Shift (A) (*n* = 126)	Fixed-Evening Shift (B) (*n* = 63)	Fixed-Night Shift (C) (*n* = 60)	F	*p*	Post-Hoc
*Physical activity*	1333.0 (1187.0)	1577.1 (1298.1)	1431.9 (1682.4)	0.69	0.502	
*Dietary behavior*	65.4 (6.9)	66.6 (7.9)	66.9 (7.6)	1.11	0.331	
General dietary behavior	60.3 (9.1)	62.8 (11.2)	65.3 (11.2)	4.95	0.008	C > A
Nutrient consumption behavior	63.7 (7.2)	63.9 (8.2)	62.6 (7.6)	0.58	0.560	
Salt consumption behavior	66.8 (15.5)	68.4 (13.3)	61.7 (16.0)	3.46	0.033	B > C
High fat-density consumption behavior	63.8 (9.4)	63.3 (9.5)	61.6 (9.9)	1.10	0.333	
Sugar consumption behavior	61.0 (10.7)	63.2 (15.2)	64.2 (14.2)	1.48	0.231	
High fiber consumption behavior	62.7 (19.3)	60.6 (20.3)	68.3 (20.9)	2.51	0.084	
*Sleep pattern*						
Sleep duration	7.4 (1.3)	8.4 (1.8)	8.4 (2.3)	10.98	<0.001	A < B, A < C
Sleep quality	7.5 (2.9)	6.7 (3.3)	7.1 (3.5)	1.30	0.274	
*Perceived Stress*	19.2 (3.8)	18.5 (5.3)	18.2 (4.1)	1.27	0.283	

*Note*: Data were presented as mean (SD); Physical activity (*MET*-*min*/*week*); Sleep duration (*hour*); *p* values were from the analysis of variance. Post-Hoc analyses were from Scheffe’s test.

**Table 4 ijerph-19-05235-t004:** Correlations among the length of rotating shift work and lifestyle patterns.

Variables	Length of Rotating Shift Work	
	*r*	*p*
*Physical activity*	0.015	0.792
*Dietary behavior*	0.149	0.008
*Sleep pattern*		
Sleep duration	−0.169	0.003
Sleep quality	0.033	0.562
*Perceived Stress*	−0.238	<0.001

*Note*: Physical activity (*MET*-*min*/*week*); Daily sleep time (*hour*); Length of rotating shift work (*year*); *p* values were from Pearson Correlation analyses. *r*: Pearson correlation coefficient.

**Table 5 ijerph-19-05235-t005:** Associations between work schedule characteristics and lifestyle patterns via multiple linear regressions.

Dependent Variables Independent Variables	Physical Activity		Dietary Behavior		Sleep Duration		Sleep Quality		Perceived Stress	
	β	*p*	β	*p*	β	*p*	β	*p*	β	*p*
*Work* *schedule type (N = 329)*										
Fixed-day shift (*n* = 80)	Reference group									
Rotating shift (*n* = 249)	0.01	0.810	0.02	0.739	0.18	0.001	0.03	0.543	0.15	0.007
*Rotating shift type (N = 249)*										
Non-fixed-rotating shift (*n* = 126)	Reference group									
Fixed-evening shift (*n* = 63)	0.06	0.361	0.06	0.416	0.27	<0.001	−0.11	0.099	−0.04	0.545
Fixed-night shift (*n* = 60)	0.01	0.883	0.10	0.156	0.25	<0.001	−0.06	0.405	−0.10	0.149
*Length of rotating shift work*	0.02	0.792	0.15	0.008	−0.17	0.003	−0.17	0.003	−0.24	<0.001

*Note*: *p* values were from multilinear regression models adjusted for covariates including age, gender, education, marriage, work unit/specialty area, length of work.

## Data Availability

Data were available with the corresponding author.

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
