# Peer review of "Impact of Rotating Shifts on Lifestyle Patterns and Perceived Stress among Nurses: A Cross-Sectional Study"

_ijerph, 2022, doi:10.3390/ijerph19095235_

Round 1

Reviewer 1 Report

Thank you for the opportunity to review this manuscript. The authors present an important and often overlooked topic regarding the associations between shift work, lifestyle factors, and perceived stress among hospital nurses. A few areas of the manuscript could use some clarification.

Methods

  • Could you provide additional information regarding recruitment and screening? For the 1,700 hospital nurses were they screened using employee records or did they contact the study coordinator and complete screening?
  • It is not clear to me the purpose of the random sampling. Could you provide some additional detail for why you selected this approach?
  • How did you get from 1,700 to 340 participants? Did everyone invited for further assessment agree to participate?
  • Did participants complete paper questionnaires? How were they returned to the researchers?

Results

  • Table 1 – for the marriage variable, is the n and % for those who are married or those who are single?
  • Table 1 – it would be helpful to include the Mean (SD) in a separate column than the n (%) to assist with readability
  • Table 1 – the methods talk about 2 main shift work types, but then table 1 includes 3 main shift work types. It would be helpful if there was a way to clarify the shift work paragraph in the methods.

Discussion

  • Lines 290-291 discuss how shift work was associated with dietary behaviors. However, the results do not demonstrate that dietary behaviors are significantly associated with shift work; the only significant relationship reported in the tables for dietary behaviors was with length of rotating shift work. Could you please clarify which results lines 290-291 are referring to?

Author Response

Your encouragement is highly appreciated. We are grateful for the opportunity to revise this manuscript. Your valuable comments are well taken and have been very helpful to the clarity of our manuscript.

Reviewer 2 Report

The high-stress levels documented among health personnel result in negative outcomes including mental health problems. Yet few studies have assessed the nature of work in relation to the negative consequences. That is why the current study that focused on the working hours of nurses and the resultant psychological problems provides a good starting point if effective interventions are to be designed.

The manuscript was generally well written.

The abstract was written according to the IMRaD format. The measures that were included in the questionnaire including the perceived stress scale can be deleted from the abstract. Add a simple conclusion to the abstract.

In the introduction, the sex variations with the variables could be explored further given that the nursing profession is generally dominated by females.

The methods include the program that was used to analyze the data.

The sample selection was done using computer-generated random numbers. Include the program that was used for this purpose.

The measures versions were from different languages including Mandarin, Taiwanese and Chinese. Could the authors comment on the final language of the questionnaire?

Why were participants with cancer excluded from the study?

The dietary behavior scale had a subscale but these were not included in the data analyses.

Any preliminary analyses that were done before the inferential statistics were performed?

Table 1 results should be presented in full such as gender and marital status.

Table 2 includes the effect size such as Cohen d for the significant results.

Table 4 results should be interpreted based on the direction of the correlation results. This should be done especially for the negative correlations between the variables.  

The sample sizes displayed in Table 2-3 were varied. This could be analyzed using nonparametric statistics instead of parametric statistics.

Explain more about the assumptions of the regression analysis in relation to the fact that some data were categorical (see Table 5).

Author Response

We are grateful for the opportunity to revise this manuscript. Your valuable comments are well taken and have been very helpful to the clarity of our manuscript.

Reviewer 3 Report

Thank you for the opportunity to review this interesting manuscript. This paper has a relevant focus for the target population and is of a great value for future research and interventions. The research has a simple and direct design to meet the research questions.

The Introduction clearly addresses the topic. It is concise, flows well from section to section, and has a clear thesis statement and aims.

The Methods and Results section are also clear and well organized.

The discussion section is also well organized and reads very well. Although the authors have mentioned some implications, it is my suggestion to highlight the clinical implications of this study, and consequently provide suggestions to improve the problem – this could be an added value for changing health/work policies.

Congratulations for your work, I have no remarks to suggest. I wish you the best for your work. 

Author Response

Your encouragement is highly appreciated. We are grateful for the opportunity to revise this manuscript.
